# The Effects of Castration, Implant Protocol, and Supplementation of *Bos indicus*-Influenced Beef Cattle under Tropical Savanna Conditions on Growth Performance, Carcass Characteristics, and Meat Quality

**DOI:** 10.3390/ani12030366

**Published:** 2022-02-02

**Authors:** Nelson Huerta-Leidenz, Nancy Jerez-Timaure, Argenis Rodas-González, Jhones Onorino Sarturi, Mindy M. Brashears, Markus F. Miller, Michel Todd Brashears

**Affiliations:** 1International Center for Food Industry Excellence, Department of Animal and Food Sciences, Texas Tech University, Lubbock, TX 79409-214, USA; nelson.huerta@ttu.edu (N.H.-L.); j.sarturi@ttu.edu (J.O.S.); Mindy.Brashears@ttu.edu (M.M.B.); mfmrraider@aol.com (M.F.M.); todd.brashears@ttu.edu (M.T.B.); 2Facultad de Agronomía, Departamento de Zootecnia, Universidad del Zulia, Maracaibo 4005, Venezuela; 3Facultad de Ciencias Veterinarias, Instituto de Ciencia Animal, Universidad Austral de Chile, Valdivia 5090000, Chile; 4Department of Animal Science, Faculty of Agricultural & Food Sciences, University of Manitoba, Winnipeg, MB R3T 2N2, Canada; Argenis.RodasGonzalez@umanitoba.ca

**Keywords:** beef, bull, grass-fed beef, carcass, cutability, palatability

## Abstract

**Simple Summary:**

Ranchers in tropical savannas must alter management to deal with forage shortages during seasonal droughts while adding value to their calf crop. Castration, pasture supplementation, and growth-promoting implant protocol (GPIP) can improve growth performance, carcass yield, and perhaps meat quality, potentially increasing profitability. The researchers measured the effects of pasture supplementation and GPIP use in bulls (Experiment I) and/or steers (Experiment II). Bulls offered a strategic protein-energy supplementation (SS) outperformed those offered a traditional mineral mix (MS) in live weight gains, carcass yield, and yield of expensive meat cuts. A combination of a high-potency implant (HPI) followed by a low-potency implant (LPI) resulted in more tender steaks from SS bulls. Tenderness was improved by HPI-LPI in SS bulls or by LPI in SS steers. It is concluded that (a) SS improved growth performance and carcass yield of bulls, while castration improved carcass quality (grading); and (b) the response to GPIP in cutability and tenderness was dependent on castration status. These results are useful for ranchers finishing calves in tropical savannas worldwide.

**Abstract:**

The effects of castration, supplementation, and implant protocol (IP) on growth, carcass characteristics, and meat quality of grass-fed cattle were evaluated. Two experiments followed a two-way ANOVA and a 2 × 2 factorial arrangement. Experiment-I, 99 bulls were evaluated for: (a) supplementation (mineral (MS) or strategic protein-energy supplementation (SS), and (b) IP (repeated (day-0 and day-90) Zeranol-72 mg implantation (Zeranol–Zeranol) or Trenbolone Acetate-140 mg/Estradiol-20 mg (day-0) followed by Zeranol-72 mg (day-90) (TBA/E2–Zeranol). Experiment II, 50 animals were evaluated for: (a) IP (like Experiment-I), and (b) male class (steers vs. bulls). In Experiment-I, SS bulls had greater growth rate, carcass yield, and yield of high-valued boneless lean cuts than MS bulls, while decreasing (*p* < 0.05) time to harvest. Steaks from SS-bulls on TBA/E2–Zeranol IP were more (*p* = 0.05) tender than SS/Zeranol–Zeranol counterparts. Experiment-II bulls had greater growth than steers, but decreased (*p* < 0.05) carcass quality aspects. Zeranol–Zeranol increased (*p* < 0.01) meat tenderness of steers. Interactions (*p* < 0.05) affected cutability (Experiment-II) and meat sensory traits (Experiment-I/II). The SS improved growth, carcass yield, and shortened days until harvest of bulls, while TBA/E2–Zeranol IP positively affected tenderness in bull meat only. Castration improved carcass quality while the implant effects on cutability and tenderness were male-class dependent.

## 1. Introduction

Non-castrated zebu-influenced calves are the main crop of cow–calf ranches operating in the savanna ecosystem of Venezuela [1]. Bull calves are usually sold at weaning to be raised under grazing systems with the expectation of compensatory body-weight gains in less harsh regions. In Venezuela, intact bull calves are preferred on grass-fed systems because of their superior growth rate on pasture and desirable yield of more muscular and leaner carcasses with additional premiums offered by federally inspected packing houses, which satisfy a robust demand by the retail sector nationwide [2]. However, castration is encouraged by the Venezuelan beef carcass grading system [3] because only castrated males (steers) under 30 months of age (MOA) are eligible for the top carcass quality grade. Producers in the region of interest (Apure State) face environmental and managerial challenges to keep their cow–calf operations sustainable, despite major investments in crossbreeding programs [4,5] and pasture infrastructure improvements [6,7]. Within grazing systems, forage quality and quantity are the major factors affecting the metabolizable protein and energy allowable for body weight gain in ruminants [8]. Several environmental and management conditions may compromise the ability of tropical pastures to supply desirable nutrients required by cattle [8]. Within those, hyperseasonality on savanna systems in which dry seasons are predominantly affecting loss of leaf mass, crude protein, and increase in senescent content of plants [8,9,10]. With the application of suitable production practices and technologies, it is possible to add value to the feeder cattle by retaining ownership of calves until terminal weight is achieved, thus improving profitability of breeding operations in the region of interest. In confined feeding operations, bulls outweigh steer counterparts in growth performance [11,12,13], but steers and bulls perform similarly under certain grazing conditions [14,15,16]. Growth enhancing technologies (steroidal and non-steroidal implants) [17,18] and strategic pasture supplementation during the dry season [10,19,20] have been proposed to shorten cattle cycles in the Apure’s neotropical savannas. Beneficial effects of single or combined steroidal implants on steer growth performance have been unanimously reported under intensive feeding [21]. Positive growth performance responses of grazing steers to zeranol, a resorcylic acid lactone with estrogenic properties (ZER) or Trenbolone acetate + Estradiol (TBA/E2) implants have also been granted positive outcomes [14,22,23,24,25,26,27,28]. However, the effects of implants on bull growth performance have been contradictory or inconsistent [14,23,28,29,30]. Potent implant combinations can have detrimental effects on steer carcass quality and beef palatability [31,32,33]; although, a meta-analysis conducted by Lean et al. [34] demonstrated that multiple-implant protocols improved beef tenderness. Nevertheless, Lean et al. [34] only included bulls in 7 out of the 181 treatment comparisons. Experiences on the combined effects of re-implant protocols and strategic supplementation on cutability (fabrication yield) and meat quality of grass-fed, tropical cattle are very rare. It was hypothesized that the combined use of implant protocols and strategic supplementation with or without castration makes feasible quality-pasture-finish of young cattle in grass-fed systems under tropical savannas conditions by positively affecting days until harvest, carcass yield, quality grade, fabrication yield of commercial cuts, and meat quality of Brahman-influenced cattle. A previous assessment was conducted in the Apure’s savannas [35] to evaluate the responses in fattening performance, carcass traits, and carcass classification/grading of grazing bulls exposed to implant protocols (ZER-ZER vs. TBA/E2-ZER), while supplementation of cultivated pastures was conducted during the dry season. Hence, a follow-up to the study by Huerta-Leidenz et al. [35] is presented herein to further explore the effects of implant protocol and pasture supplementation on fattening performance traits and cold carcass yield of bulls and examine the responses to treatments in fabrication yield and meat quality. In addition, a second experiment was designed to assess the effects of male class and implant protocol in terms of fattening, carcass performance, and meat quality.

## 2. Materials and Methods

The experiments were carried out in compliance with the guidelines of the Code of Bioethics for animal experiments of the Venezuelan National Council for Scientific and Technological Research (FONACIT) [36], which has been adopted by the Institute of Agronomical Research and is overseen by the Council for Scientific and Humanistic Development at La Universidad del Zulia (CONDES-LUZ), project protocol CONDES-LUZ # CC-0390-04.

### 2.1. Location and Pastures

The savanna ecosystem and the hydraulic infrastructure (grazing modules) in Apure State have been previously described [6,7,37], while others [1,38] have detailed the location of the experiments and its agro-climatic characteristics. Briefly, two experiments were carried out in a commercial ranch located at the Southwestern isohyperthermic savannahs of Venezuela, 25 km south of the Apure River between 7 and 8° N and between 67 and 68° W. The area corresponds to a tropical dry forest with an annual mean temperature that varies between 22 and 29 °C. The ranch has an approximate area of 25,000 ha, 8000 are flooded in the rainy months (May–October). The average rainfall is approximately 1500 mm/year, 80% of which occurs during June–October. The total area of the grazing module was 485 ha, divided into 61 paddocks of 7.4 ha of cultivated grasses. The predominant grass in the pastures was Tanner grass (*Brachiaria arrecta*), and in a lesser proportion Star grass (*Cynodon nlemfuensis*), Pará (*Brachiaria mutica*), and Alemán (*Echinochloa polystachia*). To avoid potential bias attributed to a paddock’s effect, the experimental groups were kept under rotational grazing schedule (28 d) with seven-day occupation and 21-day rest intervals, resulting in a stocking rate of 2.4 animal-unit/ha. The experiments started at the beginning of the dry season (mid-November) when most of the area becomes dry and water is restricted to a few streams, lagoons, and ponds (irrigated dry-season native and cultivated forages). The experiments were concluded in early August (rainy season). During this period the average rainfall was 92.6 mm (a minimum of 1.4 mm in January, and a maximum of 267 mm in July). The average relative humidity ranged from 69 to 75%, while the average temperature ranged from 25.7 to 28.6 °C.

The pasture nutritional composition (DM basis) was estimated on average throughout the grazing season by Huerta-Leidenz et al. [35]: average values for dry and wet seasons respectively were total digestible nutrients: 59 and 63%; crude protein: 5.1 and 8.8%; calcium: 0.27 and 0.30%; and phosphorus: 0.26 and 0.29%, respectively.

### 2.2. Animal Management

#### 2.2.1. Experiment I

Experiment I consisted of a group of 99 contemporary *Bos indicus*-influenced intact male (bull) calves with average body weight (BW) of 347.1 ± 27.92 kg and 23.0 ± 0.85 mo. of age, at the beginning of the experiment. The genetic improvement plan of the herd and overall animal management on the ranch were described in detail by Riera-Sigala et al. [1] and Plasse et al. [5]. The management scheme before and during the fattening of these bull calves had been previously described by Huerta-Leidenz et al. [35]. Briefly, the bull calves were divided into four groups, balancing by breed type, weight, and implant treatment. The experiment design was reported by Huerta-Leidenz et al. [35] and reproduced in Appendix A. Implant protocol consisted of (1) a dose of Zeranol (72 mg; Ralgro Magnum^®^ Merck & Co., Inc., Kenilworth, NJ, USA) at d-0 with re-implantation of the same dose at d-90 (ZER-ZER); or (2) a treatment consisting of a first dose containing a combo implant 140 mg trenbolone acetate + 20 mg estradiol 17β (Revalor^®^ Merck & Co., Inc., Kenilworth, NJ, USA) at d-0, followed by a re-implantation with Zeranol (72 mg) at d-90 (TBA/E2-ZER).

Supplementation treatments consisted of the traditional mineral supplementation (MS) vs. a strategic supplementation (SS). The bulls within the MS group received a mineral mixture offered continuously *ad libitum*, which induced an average daily consumption of approximately 80 g/animal-daily (DM basis). This mineral supplement contained P and Ca and other macro and microelements to complement the mineral contribution of the pasture (Appendix A), a means to meet or safely exceed the beef cattle nutrient requirements [39]. The cattle within the SS group were individually fed (once-daily) with a supplement (1 kg/d, as-fed basis) containing hydrolyzed feather meal, cane molasses, rice polish, a mineral premix with P and Ca, and 150 mg/animal-daily of sodium salinomycin(Salocin, Posistac; Phibro Animal Health Colombia SAS Bogota, DC, Colombia) during d-0 to 60 (Strategic Supplement-Phase 1; Appendix A). After d-60, animals were offered a finishing supplement (Strategic Supplement-Phase 2; Appendix A) consisting of 50% of the whole cottonseed, 27% of rice polish, 7% of minerals, 10% of feather meal, 5% of molasses, as well as encapsulated bypass fat: ether extract (EE): 22.4%, as well as an additional source of bypass protein with low ruminal degradability [35].

The second supplement was maintained for 122 d until the first heavy rainfall of the wet season. The BW at the start and end of the fattening period was determined by using an electronic, single-animal scale (Fairbank^®^ model FB2255; Fairbanks Scales Inc., Overland Park, KS, USA). Animals were sent to harvest when reaching a satisfactory conformation (thickly fleshed, well-rounded silhouettes at hindquarters, loins, and shoulders), indicating market readiness. The live animal conformation was determined by the visual evaluation of three trained personnel once a BW of 475 kg of weight was met or exceeded [35]. The average shipment BW for transportation to the harvest plant was 509.51 ± 31.70 kg [35]. Adjusted final BW was calculated by dividing the hot carcass weight (HCW) by the average dressing percent across treatments and adjusted by a 4% shrink. Carcass-adjusted average daily gain (ADG) was calculated from carcass-adjusted final BW, initial BW, and days on feed.

#### 2.2.2. Experiment II

A randomly selected group of 50 Brahman-influenced male calves (average initial BW of 342.78 ± 26.54 kg and 22.5 ± 0.1 mo. of age) was used for this grazing experiment in which all animals were offered SS pasture supplementation as described in Experiment I. The experiment design is depicted in Table 1.

Nineteen calves had been castrated at weaning (7 months of age approximately), and 31 calves were left intact (bulls). During castration, calves were handled as not to suffer unusual stress and pain as required by the code of bioethics of FONACIT [36]. Animals were immobilized in a squeeze chute, and surgical castration was performed quickly by an experienced veterinarian who supervised pre- and post-surgery treatments for preventing infections and speeding up the healing process. The ranch’s protocol for disinfection procedures and veterinary treatment before and after castration was described by Huerta-Leidenz et al. [28]. Castrated (steer) and intact (bull) calves were treated against ectoparasites and endoparasites with Ivermectin (1 mL/kg of BW; Ivermetopp Dorado, Topp Laboratories, Caracas, Venezuela) subcutaneously injected at the recommended dose according to their BW. Steers and bulls were divided into two groups, balancing by breed type and implant treatment (Table 1). Muscle thickness (1 = very heavy muscled, and 5 = lightly muscled) and frame size scores (1 = very large, and 5 = very small) were assigned to each animal at the start of the experiment following the criteria of Venezuela Decreto 181 [40].

The two-implant protocol treatments were identical to Experiment I [35]. Body weights were recorded on d-0 and when supplementation ceased on d-163. As performed by Huerta-Leidenz et al. [35], lots of finished cattle were successively sent to harvest upon reaching the same endpoint criteria of Experiment I [35]. Hip height and BW were measured at the time of shipment to the abattoir. Adjusted BW was calculated by dividing HCW by the average dressing percent across treatments and adjusted by a 4% shrink. The carcass-adjusted average daily gain was calculated from carcass-adjusted final BW, initial BW, and days on feed.

### 2.3. Harvest and Carcass Evaluation

Lots of cattle were successively slaughtered at a federally inspected abattoir located nearby the city of Barquisimeto, Lara State, approximately 500 km from the ranch. Harvest procedures and post-mortem inspections were carried out according to the standards of the Venezuelan Council of Industry Standards [41,42]

#### 2.3.1. Experiment I

Cold (48 h *postmortem*) carcass weights were recorded and dressing yields (%) were estimated with the final live weight on the supplementation test. Other carcass evaluations and the carcass grading performance of experimental groups were previously reported by Huerta-Leidenz et al. [35].

#### 2.3.2. Experiment II

The carcass evaluation has been described in detail by Huerta-Leidenz et al. [35] and Rodas-González et al. [43]. After recording the hot carcass weight to estimate the hot carcass dressing yield (%), the following traits were evaluated 48 h *postmortem* in the chilled carcass: cold carcass weight to estimate cold carcass dressing yield (%), conformation profile score (1 = Very convex, 2 = Convex, 3 = Straight, 4 = Concave, 5 = Very concave; as detailed by Huerta-Leidenz et al. [28], external fat finish (1 = Extremely abundant, 2 = Abundant, 3 = Medium, 4 = Slight, 5 = Scarce) [35], and adipose maturity (i.e., fat color, where, 1 = Ivory white, 2 = Creamy white, 3 = Light yellow, 4 = Intense yellow, 5 = Orange) according to Decreto Presidencial No. 1896 [3], ribeye area and adjusted fat thickness at the 12th-rib, degree of marbling, lean maturity (lean color and texture), and bone maturity (backbone ossification). Except for the conformation and finish scores, other traits of the chilled carcasses were evaluated according to the procedure stipulated by the USDA [44] for determining the USDA yield and (or) USDA quality grades while Venezuelan quality categories were estimated according to Decreto Presidencial No. 1896 [3].

### 2.4. Carcass Fabrication

Chilled carcass sides were reduced to subprimal/retail cuts according to the Venezuelan commercial fabrication system [41]. Experienced butchers performed this fabrication process leaving a maximum fat cover of 0.64 cm. The resulting products (cuts) and co-products (clean bone and trimmable fat) were individually weighed and the composite yield (as a percentage of the cold carcass weight) of groups of subprimal/retail cuts according to their domestic market value [43] was computed for data analysis. For Experiment I, there was missing data on the fabrication yield of the forequarter; therefore, only fabrication yield data (i.e., high-valued boneless cuts) from the hindquarter were reported. The international equivalences in the nomenclature of each commercial cut have been previously reported [43].

### 2.5. Culinary, Sensory Evaluation, and Shear Force Tests

A 20 cm long loin roast was removed from the lumbar portion, from which four steaks of 2.5 cm of thickness each were cut alternately. A pair of steaks (*longissimus dorsi lumborum*) was used for sensorial evaluation, while the other pair was used for the Warner–Bratzler shear force (WBSF) test. The excised steaks were vacuum packaged in a B620 Cryo-vac^®^ multilaminar shrink bag (Sealed Air Corp., Charlotte, NC, USA) using Koch-Ultravac^®^ packaging equipment (UltraSource LLC, Kansas City, KS, USA) and immediately frozen at −30 °C and stored at this temperature until further analyses. Culinary, sensory evaluation, and shear force protocols followed the American Meat Science Association (AMSA) guidelines [45] and have been described in detail in previous reports [28,43,46]. Briefly, cookery was performed in a Sunbeam-Oster^®^ open electric grill 1996 electric indoor grill (Sunbeam-Oster Co. Inc., Fort Lauderdale, FL, USA). Before cooking, thawed weights were recorded. Upon reaching the internal temperature of 70 °C, the steak was removed from the grill and weighed again. Cooking time was recorded, and cooking loss was estimated based on thawed and cooked steak weight. The Warner–Bratzler Shear Machine (G. R. Manufacturing, Inc., Trussville, AL, USA) was used for measuring the meat shear force according to AMSA [45]. The taste panel consisted of five to eight, 25- to 45-year-old highly trained personnel from both sexes with different levels of instruction, who tasted about 12 samples per day. Each panelist was given two or three cooked samples to score based on an 8-point structured scale for muscle fiber tenderness, amount of connective tissue, overall tenderness, juiciness, and flavor intensity (where, 1 = extremely tough, an abundant amount of connective tissue, extremely tough, extremely dry, and extremely bland, respectively; and 8 = extremely tender, no connective tissue, extremely tender, extremely juicy, and extremely intense, respectively).

### 2.6. Statistical Analyses

The Shapiro–Wilk normality test [47] was performed for each response variable. Once the conventional assumptions were fulfilled, the analysis of variance (ANOVA) was performed with the R software [48]. For bull data collected in Experiment I, a linear mixed model (LMM) was applied to a completely randomized design with a 2 × 2 factorial arrangement of treatments that included, as fixed effects, supplementation treatment, implant protocol, and the first-order interaction for the growth performance, fabrication yield, WBSF, and cookery traits. The breed type and date of shipment were included in the model as random effects. The experimental unit was the animal or the carcass. The frequency distribution of harvest lots of bulls with different fattening days, by treatment, was computed and subjected to chi-square analysis. Frequency values were compared using the chi-square option of R Core Team [48] with a significance level of 0.05. For sensory traits variables, LMM was used and panelist was included as an additional random variable. Multiple mean comparisons were made by using the Tukey–Kramer test for unbalanced data [49] with a significance level of 0.05.

Similar statistical approaches and procedures were performed with Experiment II data, following a 2 × 2 factorial arrangement of treatments with male class and implant protocol as fixed effects and first-order interactions. Breed type and shipment dates were included as random effects, while BW at the start of the trial was included as a covariate for growth and carcass traits, where it represented a significant (*p* < 0.05) source of variation for male class.

Analyses of the frequency distribution of harvest lots with different fattening days and the carcass grading performance according to the Venezuelan and USDA grading systems [3,44] were performed by male class and implant protocol. Frequency values were compared using the chi-square option of R Core Team [48] with a significance level of 0.05. As for sensory variables, the same LMM was used and panelist was included as an additional random variable. The Tukey–Kramer test (α = 0.05) [49] was used for multiple comparisons of means when the interactions were significant.

## 3. Results

### 3.1. Experiment I

#### 3.1.1. Frequency Distribution of Cattle in Harvesting Lots by Supplementation and Implant Treatments

The chi-square test detected differences (*p* < 0.01) between supplementation treatments for the frequency distribution of harvesting lots of bulls. Conversely, the distribution of these frequencies did not differ (*p* > 0.10) between implant-protocol groups. Most bulls subjected to strategic supplementation (SS) reached the desired endpoint earlier than those subjected to mineral supplementation (MS) (Table 2). By d-223 all the SS-treated bulls were off the test. In contrast, only one-third of the MS-treated bulls had a comparable growth performance in the days on feed (DOF) to their SS counterparts by d 223. The other two lots of MS-treated bulls were lagged out and had to remain on the test until reaching the endpoint at d-237 (*n* = 21) or d-258 (*n* = 17).

#### 3.1.2. Growth Traits, Carcass Dressing, and Yield of Hindquarter’s Subprimal and Coproducts

No supplementation × implant interactions (*p* ≥ 0.05) were observed for adjusted final BW, adjusted ADG, hot carcass weight or carcass dressing, or percentage yield of high valued, boneless, lean cuts (HVBLC) and derived co-products (clean bone and trimmable fat) from the carcass hindquarter (Table 3).

##### Pasture Supplementation and Implant Protocol Main Effects

Adjusted ADG was affected (*p* < 0.01) by supplementation treatment. Accordingly, the rate of gain of bulls subjected to SS outperformed (+105.8 g (+8.02%); *p* < 0.01) that of MS-treated bulls (Table 3). Moreover, SS increased the percentage of cold carcass dressing (+1.49 percentage units; *p* < 0.01) and yield of high-valued, boneless cuts (+2.04 percentage units; *p* < 0.05) compared to their MS counterparts.

Neither growth performance indicators nor the fabrication yield variables varied with the implant protocol (*p* > 0.10).

#### 3.1.3. Cookery and Quality Traits

##### Interaction Effects

Interactions of SUPPL × IMPL (*p* = 0.02) for muscle fiber tenderness, overall tenderness (*p* = 0.02), and amount of connective tissue (*p <* 0.01) were observed (Table 4). Within the MS group, these tenderness-related sensory traits did not (*p* > 0.05) vary by IMPL, whereas SS offered bulls implanted with the TBA/E2-ZER produced steaks that were perceived by the sensory panel as more tender and with a lesser amount of connective tissue than bulls offered SS and implanted with ZER-ZER and MS counterparts (*p* < 0.05; Figure 1).

No differences (*p* > 0.05) in sensory traits were detected between the SS and MS groups implanted with ZER-ZER, but within the TBA/E2-ZER implanted group, steaks from SS bullocks received greater panelist ratings for muscle tenderness, overall tenderness, and amount of connective tissue than their MS counterparts (*p* ≤ 0.05).

##### Pasture Supplementation and Implant Protocol Main Effects

The ANOVA indicated independent effects of SUPPL on cooking loss, WBSF values, and ratings for the juiciness of loin steaks (*p* < 0.01; Table 4). In turn, IMPL independently affected their cooking time, WBSF values, and juiciness ratings (*p* < 0.01).

Steaks from the SS bulls exhibited greater cooking losses, required lower force to be sheared, and were rated greater in juiciness than those from the MS group (*p* < 0.01). On the other hand, steaks from bulls implanted with the TBA/E2-ZER required longer cooking time to reach the internal temperature endpoint, had lesser WBSF values, and were rated greater in juiciness than those from the ZER-ZER group (*p* < 0.01).

### 3.2. Experiment II

#### 3.2.1. Frequency Distribution of Cattle in Harvesting Lots by Male Class and Implant Protocol

The chi-square test did not (*p* > 0.10) detect differences between male classes or implant protocols for their frequency distribution in harvesting lots (Table 5).

#### 3.2.2. Growth Performance Traits

No first-order interaction or implant protocol main effects were detected (*p* ≥ 0.06) for any growth trait (Table 6). Contrastingly, male class affected ADG2, BW at end of supplementation, final BW at shipment, adjusted BW at shipment, and chronological age of cattle.

Compared to steers, bulls grew at a slightly faster rate of gain (40 g/d; *p* < 0.05) from d-0 to time of shipment to the abattoir and, hence, were 40 kg and 27 kg heavier in BW at the end of the supplementation test (*p* < 0.05) and at the time of shipment to harvest (*p* < 0.01), respectively. Nonetheless, at harvest steers were younger (26 days approximately) than bulls (*p* < 0.01).

#### 3.2.3. Carcass Traits

No first-order interaction was detected for any of the carcass traits under study (*p* ≥ 0.10). Mean comparisons of carcass traits according to male class or implant protocol are depicted in Table 7.

Few differences in carcass traits were detected between steers and bulls. Bull carcasses were approximately 17 kg heavier, exhibited thinner backfat with wider thighs, less favorable conformation scores, and lower marbling levels compared to steers (*p* < 0.05; Table 7).

#### 3.2.4. Frequency Distribution of Carcass Categories/Grades According to Male Class and Implant Protocol

The chi-square test did not (*p* > 0.05) detect differences between implant protocols in the frequencies for Venezuelan quality categories or USDA quality/yield grades. Instead, the male class affected the frequency distribution of carcasses in the Venezuelan and US quality categories/grades. Table 8 shows the distribution of such frequencies.

In the group of steers, 8 out of 19 carcasses (42.1%) reached the AA category, the top-quality Venezuelan grade, which is not accessible for bulls [3]. In contrast, a comparatively lower percentage (9.7%) of bulls reached the top-quality Venezuelan category (“A”) for which this intact male class is eligible [3]. Instead, most bull carcasses (68%) categorized “B”, the second Venezuelan quality category.

Regarding the USDA carcass grading performance, only 21% (*n* = 4) of the steer carcasses reached the Select grade, which was unreachable by any carcass from the bull group. More than half of the steer carcasses (52.6%) were graded high Standard while 48.4% of those derived from intact males were graded as low Standard “bullocks”, a designation reserved for young bulls under 30 months of age [44]. Twelve out of 31 intact males (38.7%) were classified as “bulls” because they exhibited a B maturity level and, hence, were not eligible for any quality grade [44].

Although no effects on the distribution frequency of USDA yield-grade carcasses were detected, there was a trend (*p* = 0.12) of bulls to perform better than steers as denoted by a numerical greater percentage (+16.8%) of carcasses graded with the superior USDA Yield grade 1.

#### 3.2.5. Carcass Yield in Subprimal Cuts and Coproducts

A male class × IMPL was detected (*p* = 0.01) for percentage yield of medium-valued cuts, percentage of total cuts, and percentage of trimmable fat (Table 9)**.** Steers implanted with ZER-ZER had lesser yields of medium-valued cuts and total cuts and resulted in a greater proportion of trimmable fat as compared with the ZER-ZER treated bulls (*p* < 0.05). In the group of steer carcasses, a greater (*p* < 0.01) proportion of fat was trimmed from those treated with ZER-ZER compared to their TBA/E2-ZER counterparts (Table 9, Figure 2).

#### 3.2.6. Cookery and Meat Quality Traits

A male class x IMPL interaction (*p* < 0.01) was observed for WBSF, ratings for muscle fiber tenderness, amount of connective tissue, and overall tenderness (Table 10, Figure 3).

Steaks derived from ZER-ZER-treated steers required lesser WBSF (*p* < 0.01) and were rated greater by the panelists for muscle fiber tenderness, overall tenderness, and amount of connective tissue (*p* < 0.01) as compared to those derived from TBA/E2-ZER-treated steers. Moreover, steaks from ZER-ZER-treated steers were rated as more tender with a lesser amount of connective tissue than their ZER-ZER bull counterparts (*p* < 0.01). On the contrary, steaks from TBA/E2-ZER-treated steers required more shear force and were rated lower for muscle fiber tenderness, overall tenderness, and amount of connective tissue (*p* < 0.01) with respect to TBA/E2-ZER-treated bulls (*p* > 0.01). In turn, steaks from TBA/E2-ZER-treated bulls exhibited greater ratings in muscle fiber and overall tenderness and were perceived with a lesser amount of connective tissue than those from the ZER-ZER-treated bulls (*p* < 0.01).

##### Male Class and Implant Protocol-Independent Effects

The ANOVA detected effects of male class and implant protocol on cooking loss as well as an effect of male class on flavor intensity (*p* < 0.05; Table 10). Steaks from steers and those from groups implanted with the TBA/E2-ZER protocol had lesser cooking losses than their bull and ZER-ZER counterparts, respectively (*p* < 0.05). On the other hand, regardless of the implant protocol, steaks from steers rated greater in flavor intensity (*p* = 0.02) and tended to be perceived as with greater numerical juiciness (*p* = 0.05) than those from bulls.

## 4. Discussion

### 4.1. General

Most of the comparisons of pasture supplementation treatments and/or implant agents in the South American tropics focused on growth performance traits and very few carcass traits if any (e.g., weights and dressing percentages) [14,17,18,22,23,24,25,26,27,30,50,51,52,53].

Regarding the present study, it is noteworthy that the groups in both experiments were well-balanced by breed type (Appendix A and Table 1) with a predominance of crossbreds. Regarding initial BW only in Experiment II, a significant variation by male class on the initial BW was detected (Table 6), and hence the model was adjusted for growth and carcass traits at a constant initial BW. Additionally, Experiment II did not show differences in either muscle thickness score or frame size score between treatments of cattle at the start of the fattening test. This equalization is important because Connell et al. [51] with similar livestock in the same ranch had reported the frame size score was a meaningful source of variation in final BW in the supplemented group of cattle.

### 4.2. Grass-Fed Finishing Performance

Implant protocols were comparable in eliciting animal growth response. Neither experiment showed effects of implant strategy on the frequency distribution of harvest lots, adjusted ADG, adjusted final BW, or chronological age at slaughter. This observation had been reported with bulls in Experiment I by Huerta-Leidenz et al. [35] based on interim (non-adjusted) ADG or BW. The enhancing effects of strategic supplementation on adjusted ADG of bulls were clear in Experiment I, where it expedited shipment of finished cattle to harvest. Positive use of strategic supplementation on growth traits has been previously reported in different tropical environments with zebu-type cattle [54,55].

Regarding the male class, there is considerable scientific evidence to substantiate the biological advantage in the growth performance of bulls over steers under intensive feeding conditions [11,12,13]. However, under the grass-feeding conditions of the American tropics, with the exception of Rubio and Montiel [30], other researchers [14,15,16] did not observe differences in final BW between non-implanted grass-fed steers and bulls.

In Experiment II, the more advantageous performance (BW at end of supplementation, final BW on shipment, adjusted BW, and non-adjusted ADG from d0 to d of shipment) of bulls over the steers may be explained because both male classes were aggressively implanted and strategically supplemented. Kept under very similar experimental conditions to those of Experiment II, non-implanted bulls also exhibited significantly faster rates of gain than non-implanted steers; however, the inverse response was observed when both steers and bulls received a single implant of TBA/E2 [28]. The more clear advantage in adjusted BW of implanted bulls vs. implanted steers in Experiment II (+5.42%; *p* < 0.01) compared to the non-significant difference in favor of bulls vs. steers (+2.62%; *p* = 0.32) observed under very similar management in Huerta-Leidenz et al. [28] might be due to the fact that all experimental cattle in Experiment II were strategically supplemented, with a greater quality (protein and energy) supplement than the poultry-litter based supplement used in the Huerta-Leidenz et al. [28] research. In this regard, Rodríguez et al. [16] have pointed out that energy requirements for the maintenance of bulls are greater than steers; consequently, if grass-feeding does not fulfill the dietary needs of bulls for anabolic functions, the two male classes can grow at a similar rate, particularly under hot tropic environments that compromise the utilization of low-quality forages and demand an increased metabolic rate for heat dissipation. Furthermore, the predominance of crossbred cattle in Experiment II vs. the use of high-grade Brahman/Zebu cattle in the report of Huerta-Leidenz et al. [28] should be highlighted. Plasse et al. [5] postulated that under favorable environmental conditions of the Apure savanna (i.e., improved pasture and feeding conditions) crossbreds can better express their genetic advantage vs. purebred zebu cattle. Therefore, the presumable lack of heterosis in high cross Zebu cattle combined with the inferior quality of the supplement could influence the more attenuated response of bulls to TBA/E2 implant in our previous research [28].

### 4.3. Carcass Performance

Worldwide, tropical, *Bos indicus*-influenced cattle are typically slaughtered over 3 years of age, producing lean, low-yielding carcasses [56]. In both experiments, the average chronological age of grass-finished cattle required to reach the main endpoint criteria (a satisfactory conformation) for the market was less than 30 months, which represents a noticeable achievement compared to the typical stocker-to-finish operations in the country. A survey of grass-fed harvested cattle in Venezuela [57] reported a mean BW at harvest of 465 ± 19.0 kg at > 36 months of age (by dentition) while in Colombia Flórez et al. [58] reported lighter BW at harvest in the range of 453.4–463.63 kg at 31.3–32.6 months of age by dentition in a sample of Zebu and Zebu × *B. taurus* (Criollo breeds) harvested cattle.

Strategic supplementation of bulls in Experiment I improved cold carcass dressing maintaining practically the same advantage (1.49 percentage points) detected previously [35] for hot carcass dressing (+1.5%) over their MS counterparts. In the latter study [35], the strategic supplementation not only increased hot carcass dressing but also produced bull carcasses that tended to have a numerically thicker backfat (*p* = 0.07) with a younger skeletal maturity. This somewhat agrees with Jerez-Timaure and Huerta-Leidenz [59] who supplemented bulls with a poultry litter-based ration on the same ranch. However, poultry litter supplementation reduced the yield of high-value boneless cuts (HVBC) [59]. These noticeable differences in the commercial composition of carcasses may be partially related to the protein/energy utilization when extra ruminal degradable protein is offered to ruminant [60], such as in the case when poultry-litter was supplemented [59]. Although most positive effects regarding protein supplementation occur when dietary CP intakes are low (increasing intake and ruminal organic matter degradability); the opposite, represented by high levels of ruminal degradable protein, may have negative effects on animal energy use, which may negatively affect adipocytes or production efficiency [61]. In the current experiments (I and II), the low amount of ruminal degradable protein of supplements (phase 1 and 2) represented by feather meal (RDP = 30% of the CP) and whole cottonseed (RDP = 60% of the CP) combined with the low-supplementation level (1 kg/animal-daily) perhaps were not enough to induce a decrease in energy utilization efficiency, deposition of adipocytes, and consequently HVBC.

In Experiment II, it is interesting that steers were harvested at a lighter adjusted BW (*p* < 0.01) than bulls but at a younger chronological age (*p* < 0.01), suggesting that the desirable conformation (main endpoint) was reached earlier by steers probably because fatness positively influences the judging of body conformation (shape) [62]. This may explain why steers tended to require a numerically shorter time of fattening (*p* = 0.07) than bulls. The appreciation of a better conformation score in the steer carcasses is in line with this possibility. Moreover, marbling scores were in favor of the steer carcasses. It should be noted that in Experiment II, all cattle were offered strategic supplementation; thus, this practice could assist in promoting the few differences in carcass quality detected between male classes. Although more desirable scores in conformation and marbling were assigned to carcasses derived from steers, these were small in magnitude and might be considered irrelevant. However, the greater grading performance (Table 8) of steers vs. bulls according to both Venezuelan and US systems should not be disregarded. Despite reaching a very young (A) maturity that makes them eligible for USDA quality grades as “bullocks”, the Brahman-influenced bulls finished on pasture do not usually grade Prime or Choice due to severe deficiencies in marbling levels. These bull carcasses did not surpass “slight” amounts [59] or “practically devoid” of marbling [63], resulting in USDA Standard or Select grades. With a more advanced (B) maturity, traces of marbling have been reported in carcasses from four genetic groups (zebu and zebu x dairy crossbreds) of bulls implanted with a single dose of TBA/E2 and fed during 147 d under feedlot conditions [64]. Few studies have reported the quality grading performance of *Bos indicus*-influenced steer carcasses finished on tropical pastures with or without supplementation. In Experiment II, the group of steers exhibited a better performance in terms of Venezuelan quality category than in a previous comparison [28] because 42% of their carcasses graded AA (the top-quality category in Venezuela) while none of the steers in the previous study [28] achieved that level of quality. Nevertheless, steers in both studies performed similarly in USDA quality grade with a predominance (>70%) of USDA Standard carcasses.

Steers that received the high-potency implant strategy (TBA/E2-ZER) had greater leanness compared to their ZER-ZER-implanted steer counterparts. Such effect is depicted by the class × implant protocol interaction (Experiment II, Figure 2), in which less trimmed fat (waste) upon fabrication was observed for those carcasses. Given bulls and steers treated with TBA/E2-ZER did not differ in their proportions of medium-valued cuts, total cuts, or trim fat, it is plausible to speculate that the ZER-ZER protocol could increase leanness of bull carcasses and further reduce the expected lower proportion of trimmable fat with a greater yield in medium-valued cuts (composed mostly of muscles from the chuck) compared to steers [62]. The greater percentage yields of bulls vs. steers in medium-valued cuts are expected due to differential patterns of muscle growth between the two male classes. Most of the cervical muscles and the scapular belt have greater development in bulls, due to gonadal influences [62]. The same differential in carcass composition has been observed when bull carcasses are compared to those from cow and heifer counterparts [2]. The combination of a lower percentage of trimmed fat and a greater percentage yield of medium-valued cuts in ZER-ZER-implanted bulls could favor their greater percentage yield in total cuts compared to ZER-ZER steers.

### 4.4. Eating Quality and Cookery Traits

Publications addressing the effect of supplementation strategies and/or implant protocol on the sensory and textural quality of beef produced under tropical grazing conditions are very scarce.

#### 4.4.1. Effects of Pasture Supplementation

According to previous experiments [28,59,65,66,67], the type of supplementation (ingredients/protein:energy ratio) during grazing of tropical pasture affects the outcome in terms of beef palatability. The supplement used herein based on feather flour, rice flour, and whole cottonseed (Experiment I) resulted in bull steaks with greater cooking losses, lower WBSF, and greater juiciness ratings than those from the MS group, whereas the tenderness-related sensory traits (muscle fiber tenderness, overall tenderness, and amount of connective tissue) of bull steaks were affected by the SUPPL × IMPL interaction (to be discussed later in the implant discussion). In two previous studies [28,59], carried out in the same ranch with a mixture of poultry litter and rice polishing offered to Brahman-influenced bulls, the traits related to meat tenderness (WBSF, ratings for amount of connective tissue and tenderness) slightly worsened with respect to the control (MS) group.

Acosta Castellanos [65] in Colombia, reported that beef from cattle-fed forage or forage + grain diets were not different in WBSF (6.60 and 7.20 kg, respectively). Unexpectedly, cattle offered vegetable residues (vegetables, fruits, and tubers) resulted in greater WBSF values (8.35 kg) than cattle offered forage only [65].

In the temperate zones, the discussion has been focused on grass- vs grain-fed beef in confined or semi-confined conditions, which are not particularly useful for producers in tropical rangeland environments. However, few studies have explored finishing cattle under pasture supplementation strategies. In Germany, Schmutz et al. [66] fattened steers using two different grazing systems (continuous vs rotational grazing systems) supplemented with medium (9.7% CP; 1.5 kg/animal-daily) or low (9.7% CP; 0.75 kg/animal-daily) concentrate levels over 93 d and reported no effects on cooking loss or WBSF in longissimus dorsi and semitendinosus muscles. In partial agreement, Duynisveld et al. [67] in Canada, compared steers finished on a silage/barley totally mixed ration (TMR) vs. pasture either with no supplement or supplemented with 5 kg barley or 2 kg whole roasted soybeans, and no differences were found in shear force and sensory characteristics, but cooking losses were greater in beef steaks from TMR and non-supplemented pasture than those from barley- or soybean-supplemented pasture.

#### 4.4.2. Effects of Implants

Implants are used routinely by U.S. beef producers to increase the rate and efficiency of growth in steers and heifers, from suckling to finishing phases of production; however, “aggressive” and/or repetitive use of implants may be detrimental to beef carcass quality and/or tenderness [31,32,33,68,69]. The production of finished bulls with or without anabolic implants is not a common practice in the U.S.; in contrast, bull production prevails in Venezuela and other countries of tropical America [56] but with limited use of anabolic implants. It is generally accepted [11,12,43] that bull steaks are consistently tougher when compared with steers or heifers at the same age. However, several studies have indicated the effect of the implant on meat quality depends on the male class. Hunt et al. [70] reported WBSF of LM steaks from bulls implanted solely with Trenbolone acetate (TBA, 120 mg) or TBA/E2 implanted once at the start of the experiment were comparable to those from non-implanted and implanted steers. Moreover, the TBA/E2 increased sensory panel tenderness and connective tissue ratings in steaks from bulls compared with steaks from non-implanted bulls and bulls implanted with TBA alone [70]. According to these researchers [70], the promising results in quality enhancement of bull meat by implanting TBA/E2 could be due to an increase in fatness (fat thickness and chemical intramuscular fat, which would improve palatability, comparable to beef from the non-implanted steers) [70]. In this study [70], implant treatments did not affect sensory panel scores for juiciness or flavor of steaks from bulls or steers; an observation that is aligned with current results in Experiment II.

In Experiment I, the SUPPL × Implant protocol interaction showed: (a) the MS supplemented group did not elicit any variation in tenderness-related traits (muscle fiber tenderness, overall tenderness, and amount of connective tissue) of bull meat due to implant protocol; and (b) in the SS group of bulls implanted with TBA/E2-ZER, the steaks received greater ratings for muscle fiber tenderness, overall tenderness, and amount of connective tissue compared to those implanted with ZER-ZER. Moreover, SS bulls in Experiment II had a similar positive response to TBA/E2-ZER. Instead, steers were more responsive to ZER-ZER in terms of improving tenderness-related characteristics including shear force (Experiment II).

Current observations somewhat support Lean et al. [34] whose meta-analysis showed multiple implant protocols improved beef tenderness. However, current results also suggest the multi-Implant protocol that is designed to obtain this beneficial response in bull meat should be different from those that have been effective in eliciting a similar response in beef from steers.

The sensory ratings for the TBA/E2-ZER bull steaks in both experiments and for ZER-ZER steer steaks in Experiment II were below 5, and the mean WBSF value was greater than the tenderness threshold (WBSF = 4.09 kg) of Rodas-Gonzalez et al. [46]. These results indicate that the degrees of improvement in palatability traits with any of the tested implant protocols were not good enough from the consumer acceptability standpoint.

## 5. Conclusions

Ranchers in the Apure’s neotropical savannas face environmental and management challenges to keep their cow–calf operations sustainable despite major investments in cattle genetics (crossbreeding programs) and pasture infrastructure. This research tested the use of non-traditional management practices in the zone (castration, pasture supplementation, and production technologies) for developing grass-fed systems in their own premises to add value to feeder cattle. Strategic supplementation of *Bos indicus*-influenced cattle under neotropical savanna conditions was feasible and brought about meaningful improvements in the rate of gain, carcass dressing, the yield of high-valued boneless cuts of bulls plus slightly more desirable juiciness ratings, and shear force values of grass-fed bull meat. The comparison of strategically supplemented bulls vs. steer counterparts confirmed the superiority of bulls over steers in growth performance, and (b) the need for implementing castration if improvements in carcass quality are desired. The response of supplemented *Bos indicus*-influenced males to the implant protocol in cutability and tenderness-related traits were dependent on their sex class. In general, the improvements in palatability attributes observed herein for bulls and steers with individual treatments or a combination of treatments were modest and may result insufficient to please local consumers. The main limitation of the present study, particularly in Experiment I, was the low number of experimental units per treatment and the lack of a negative-control group (non-implanted) to quantify the effects of the implant protocols. Further studies with larger sample size and additional implant treatments are needed to evaluate the effects of sex class and implant protocol.

## Figures and Tables

**Figure 1 animals-12-00366-f001:**
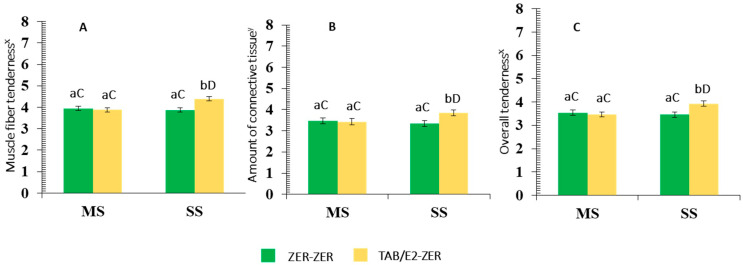
Mean values ± standard error for pasture supplementation (SUPPL) × implant protocol (IMPL) interaction (*p* = 0.02) for muscle fiber tenderness (*p* < 0.01; **A**), for connective tissue (*p* < 0.01 **B**), and overall tenderness (*p* = 0.02; **C**). Panelist ratings for bull meat (^X^ 1 = extremely tough and 8 = extremely tender. ^Y^ 1 = abundant amount of connective tissue, 8 = no connective tissue). SUPPL: SS: strategic supplementation; MS: only mineral supplementation as a control treatment. IMPL: ZER-ZER: corresponds to a dose (72 mg) of zeranol at d-0 followed by a second identical dose at d-90; TBA/E2-ZER: corresponds to a first dose containing 140 mg trenbolone acetate + 20 mg estradiol 17β at d-0, followed by a second dose of zeranol (72 mg) at d-90. Bars with a common superscript lowercase letter (a, b) for IMPL treatments within the same SUPPL treatment do not differ (*p* > 0.05). Bars with a common superscript uppercase letter (C, D) for SUPPL treatment within the same implant treatment, do not differ (*p* > 0.05).

**Figure 2 animals-12-00366-f002:**
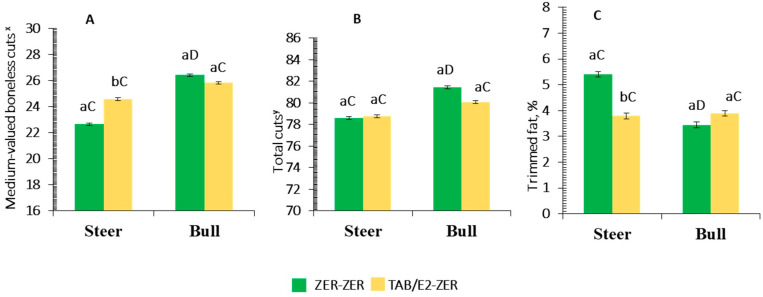
Mean values ± standard error for male class (steer, bull) × implant protocol (IMPL) interaction (*p* < 0.05) for percentage yield of cold carcass weight in medium-valued boneless cuts (^x^ sum of shoulder clod and top blade + chuck tender + chuck roll) (*p* = 0.01; **A**); total (saleable) cuts (^y^ sum of the high-, medium-, and low-valued cuts) (*p* = 0.05; **B**); and trimmed fat (*p* < 0.01; **C**). IMPL: ZER-ZER: corresponds to a dose (72 mg) of zeranol at d-0 followed by a second identical dose at d-90; TBA/E2-ZER: corresponds to a first dose containing 140 mg trenbolone acetate + 20 mg estradiol benzoate at d-0, followed by a second dose of zeranol (72 mg) at d-90. Bars with a common superscript lowercase letter (a, b) for IMPL treatments within the same male class do not differ (*p* > 0.05). Bars with a common superscript uppercase letter (C, D) for male class within the same IMPL, do not differ (*p* > 0.05).

**Figure 3 animals-12-00366-f003:**
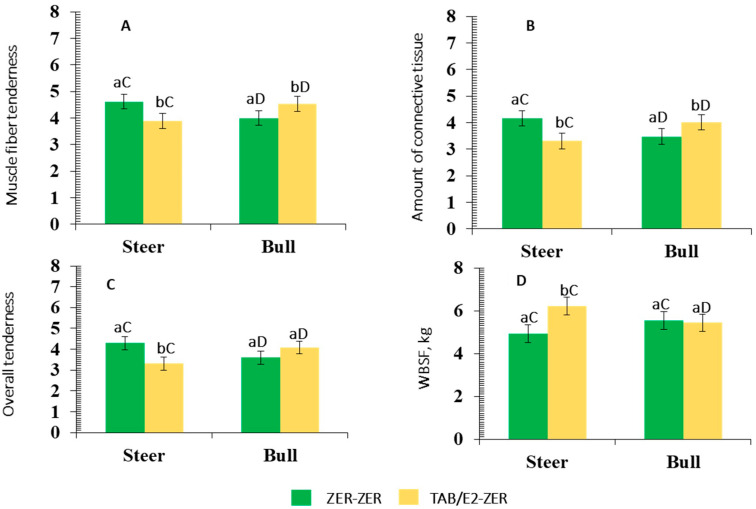
Mean values ± standard error for male class (steer, bull) × implant protocol (IMPL) interaction for muscle fiber tenderness (*p* < 0.01; **A**), amount of connective tissue (*p* < 0.01; **B**), overall tenderness (*p* < 0.01; **C**), and Warner–Bratzler shear force (WBSF, kg) (*p* < 0.01; **D**). Panelist ratings (^x^ 1 = extremely tough and 8 = extremely tender; ^Y^ 1 = abundant amount of connective tissue, 8 = no connective tissue). IMPL: ZER-ZER: corresponds to a dose (72 mg) of zeranol at d-0 followed by a second identical dose at d-90; TBA/E2-ZER: corresponds to a first dose containing 140 mg trenbolone acetate + 20 mg estradiol benzoate at d-0, followed by a second dose of zeranol (72 mg) at d-90. Bars with a common superscript lowercase letter (a, b) for IMPL treatments within the same male class do not differ (*p* > 0.05). Bars with a common superscript uppercase letter (C, D) for male classes within the same IMPL, do not differ (*p* > 0.05).

**Table 1 animals-12-00366-t001:** Experimental design indicating the distribution of observations (*n*) by breed type, male class, and implant protocol (Experiment II).

Breed Type	Steer		Bull		Total (*n*)
	ZER-ZER ^a^	TBA/E2-ZER ^b^	ZER-ZER ^a^	TBA/E2-ZER	
F1-Angus	1	2	3	3	9
F1-Romosinuano	3	2	3	3	11
F1-Senepol	2	2	4	2	10
F1-Simmental	1	2	4	3	10
Commercial Brahman	2	2	2	4	10
Total	9	10	16	15	50

^a^ ZER-ZER: corresponds to a dose (72 mg) of zeranol at d 0 followed by a second dose (72 mg) of zeranol at d-90; ^b^ TBA/E2-ZER: corresponds to a first dose containing 140 mg trenbolone acetate + 20 mg estradiol 17β at d-0, followed by a second dose (72 mg) of zeranol at d-90.

**Table 2 animals-12-00366-t002:** Frequency distribution by fattening days according to supplementation treatment and implant protocol (Experiment I).

Time until Harvest (Days)	SUPPL		Implant Protocol		Total (*n*)
	MS ^a^ *n* (%)	SS ^b^ *n* (%)	ZER-ZER ^c^ *n* (%)	TBA/E2-ZER ^d^ *n* (%)	
181	1 (7.7)	12 (92.33)	5 (38.5)	8 61.5)	13
195	0 (0)	13 (100)	7 (46.2)	6 (46.2)	13
209	5 (29.4)	12 (70.0)	10 (41.2)	7 (41.2)	17
223	13 (72.2)	5 (27.8)	7 (61.1)	11 (61.1)	18
237	21 (100)	0 (0)	12 (42.9)	9 (42.9)	21
258	17 (100)	0 (0)	9 (47.1)	8 (47.1)	17
	χ^2^ = 65.98; *p* = 0.0006		χ^2^ = 2.66; *p* = 0.75		*N* = 99

^a^ MS: mineral supplementation designed as control treatment. ^b^ SS: strategic supplementation. ^c^ ZER-ZER: corresponds to 72 mg of zeranol at d-0 followed by a second dose (72 mg) of zeranol at d-90; ^d^ TBA/E2-ZER: corresponds to first dose containing 140 mg trenbolone acetate + 20 mg estradiol 17β at d-0, followed by a second dose (72 mg) of Zeranol (ZER) at d-90.

**Table 3 animals-12-00366-t003:** Growth and fabrication yield traits of bull carcasses according to supplementation and implant treatments (Experiment I).

Variable	Supplementation (SUPPL)		Implant Protocol (IMPL)			*p*-Value		
	MS ^a^ (*n* = 57)	SS ^b^ (*n* = 42)	ZER-ZER ^c^ (*n* = 50)	TBA/E2-ZER ^d^ (*n* = 49)	SEM	SUPPL	IMPL	SUPPL × IMPL
Initial BW, kg	347.16	346.90	344.44	349.7	2.80	0.96	0.45	0.88
Adjusted final BW, kg ^e^	490.31	487.54	484.95	493.40	12.5	0.46	0.54	0.32
Adjusted ADG ^f^ (0–d of shipment), g	606.22	711.98	640.22	662.18	8.3	<0.01	0.18	0.57
Cold carcass weight, kg	288.27	294.21	289.02	292.60	1.82	0.71	0.84	0.10
Cold carcass dressing,%	56.46	57.95	57.23	56.96	0.17	<0.01	0.65	0.13
High-valued boneless cuts, % ^g^	33.04	35.06	34.62	33.16	0.31	<0.01	0.25	0.49
Total clean bone,%	7.92	7.71	7.71	7.76	0.12	0.50	0.78	0.72
Trimmed fat,%	3.48	3.80	3.53	4.02	0.16	0.83	0.78	0.91

^a^ MS: mineral supplementation as a positive control. ^b^ SS: strategic supplementation. ^c^ ZER-ZER: corresponds to a dose (72 mg) of zeranol at d-0 followed by a second dose (72 mg) of Zeranol at d-90; ^d^ TBA/E2-ZER: corresponds to a first dose containing 140 mg trenbolone acetate + 20 mg estradiol 17β at d-0, followed by a second dose (72 mg) of zeranol at d-90. ^e^ Carcass-adjusted final BW was calculated from HCW divided by the average dressing percent across treatments and adjusted by a 4% shrink. ^f^ Carcass-adjusted ADG was calculated from carcass-adjusted final BW, initial BW, and days on a feed from 0 to d of shipment. ^g^ Tenderloin + ribeye and strip-loin + Top sirloin + Eye of round + Top (inside) round + Bottom (outside) round + knuckle + Tri-tip + Heel of round.

**Table 4 animals-12-00366-t004:** Cookery traits, Warner–Bratzler shear force, and trained panel ratings of bull steaks according to supplementation and implant treatments (Experiment I).

Variable	Supplementation (SUPPL)		Implant Protocol (IMPL)			*p*-Value		
	MS ^a^ (*n* = 57)	SS ^b^ (*n* = 42)	ZER-ZER ^c^ (*n* = 50)	TBA/E2-ZER ^d^ (*n* = 49)	SEM	SUPPL	IMPL	SUPPL × IMPL
Cooking loss,%	32.44	34.18	32.86	33.48	0.29	<0.01	0.16	0.06
Cooking time, min	79.19	78.37	77.81	79.90	0.42	0.07	0.01	0.11
WBSF, kg ^e^	5.93	5.53	5.99	5.53	0.07	<0.01	<0.01	0.19
Muscle fiber tenderness ^f^	3.91	4.13	3.91	4.10	0.06	0.02	0.04	0.02
Amount of connective tissue ^g^	3.45	3.59	3.42	3.60	0.06	0.10	0.15	<0.01
Overall tenderness ^f^	3.50	3.70	3.51	3.66	0.06	<0.01	0.32	0.02
Juiciness ^h^	4.70	4.95	4.17	4.89	0.03	0.01	0.01	0.37
Flavor intensity ^i^	5.77	5.74	5.75	5.78	0.02	0.56	0.66	0.75

^a^ MS: mineral supplementation as a positive control. ^b^ SS: strategic supplementation. ^c^ ZER-ZER: corresponds to double-dose (72 mg) of zeranol at d-0 followed by a second dose (72 mg) of zeranol at d-90; ^d^ TBA/E2-ZER: corresponds to a first dose containing 140 mg trenbolone acetate + 20 mg estradiol benzoate at d-0, followed by a second dose (72 mg) of zeranol. ^e^ WBSF: Warner–Bratzler shear force expressed in kilograms. at d-90. ^f^ 8-point hedonic scale (1 = extremely tough, and 8 = extremely tender). ^g^ 8-point hedonic scale, (1 = abundant amount of connective tissue, and 8 = no connective tissue). ^h^ 8-point hedonic scale (1 = extremely dry, and 8 = extremely juicy). ^i^ 8-point hedonic scale (1 = extremely bland, and 8 = extremely intense).

**Table 5 animals-12-00366-t005:** Frequency distribution by fattening days according to male class and implant protocol (Experiment II).

Time until Harvest (Days)	Male class		Implant Protocol		Total (*n*)
	Steer *n* (%)	Bull *n* (%)	ZER-ZER ^a^ *n* (%)	TBA/E2-ZER ^b^ *n* (%)	
181	8 (46.7)	8 (53.3)	7 (46.7)	8 (53.3)	15
195	7 (41.2)	10 (58.8)	11 (64.7)	6 (35.3)	17
209	3 (30.0)	7 (70.0)	4 (40.0)	6 (60.0)	10
223	2 (28.6)	5 (71.4)	3 (42.9)	4 (57.1)	7
	χ2 = 1.06; *p* = 0.78		χ2 = 2.06; *p* = 0.56		*N* = 49

^a^ ZER-ZER: corresponds to a dose (72 mg) of zeranol at d-0 followed by a second dose (72 mg) of zeranol at d-90; ^b^ TBA/E2-ZER: corresponds to a first dose containing 140 mg trenbolone acetate + 20 mg estradiol 17β at d-0, followed by a second dose (72 mg) of zeranol at d-90.

**Table 6 animals-12-00366-t006:** Growth performance traits according to male class and implant protocol (Experiment II).

Variable	Male Class		Implant protocol			*p*-Value		
	Steer (*n* = 19)	Bull (*n* = 31)	ZER-ZER ^a^ (*n* = 25)	TBA/E2-ZER ^b^ (*n* = 25)	SEM	CLASS	IMPL	CLASS × IMPL
Initial BW, kg	333.78	348.90	339.16	346.40	3.75	0.03	0.36	0.22
Muscle thickness score ^c^	2.15	2.16	2.16	2.16	0.08	0.12	0.16	0.07
Frame size score ^d^	2.11	1.96	2.08	1.95	0.09	0.57	0.41	0.11
Hip height, cm	135.01	134.67	133.8	135.8	3.61	0.16	0.97	0.14
Chronological age, mo.	28.80	29.66	29.21	29.46	0.78	0.04	0.54	0.12
BW at end of supplementation test, kg	475.26	500.64	484.56	497.44	13.69	0.03	0.42	0.65
Final BW at shipment d ^e^, kg	484.21	511.22	494.88	507.04	13.80	0.02	0.32	0.85
ADG1 (0–180 d), g	800.29	843.01	814.43	839.11	16.77	0.22	0.41	0.71
ADG2 (0–d of shipment), g	777.63	817.57	790.71	814.07	15.75	0.04	0.12	0.27
Adjusted BW at shipment kg ^f^	464.84	490.77	475.08	486.75	14.67	<0.01	0.44	0.73
Adjusted ADG2 ^g^, g	677.74	714.42	689.89	711.07	4.75	0.18	0.25	0.45
Fattening days	195.00	199.52	197.24	198.36	5.47	0.07	0.18	0.15

^a^ ZER-ZER: corresponds to a dose (72 mg) of zeranol at d-0 followed by a second dose (72 mg) of zeranol at d-90; ^b^ TBA/E2-ZER: corresponds to a first dose containing 140 mg trenbolone acetate + 20 mg estradiol 17β at d-0, followed by a second dose (72 mg) of zeranol at d-90. ^c^ 1 = very heavy muscled, and 5 = lightly muscled [40]. ^d^ 1 = very large, and 5 = very small [40]. ^e^ Shipment day was the date of loading cattle from the ranch to the abattoir after reaching the endpoint. ^f^ Carcass-adjusted final BW was calculated from HCW divided by the average dressing percent across treatments and adjusted by a 4% shrink. ^g^ Carcass-adjusted ADG2 was calculated from carcass-adjusted final BW, initial BW, and days on feed.

**Table 7 animals-12-00366-t007:** Carcass traits according to male class and implant protocol (Experiment II).

Variables	Male Class		Implant Protocol			*p*-Value		
	Steer (*n* = 19)	Bull (*n* = 31)	ZER-ZER ^a^ (*n* = 25)	TBA/E2-ZER ^b^ (*n* = 25)	SEM	CLASS	IMPL	CLASS × IMPL
Hot carcass weight, kg	285.31	302.74	293.08	299.16	3.08	0.02	0.48	0.86
Hot carcass dressing yield,%	58.92	59.23	59.21	59.01	0.21	0.64	0.68	0.94
Cold carcass weight, kg	279.61	296.68	287.21	293.17	3.02	<0.01	0.21	0.71
Cold carcass dressing yield,%	57.74	58.04	58.03	57.93	0.20	0.48	0.63	0.95
Conformation score ^c^	3.16	3.74	3.48	3.56	0.13	<0.01	0.10	0.15
Ribeye area, cm^2^	81.70	85.58	86.11	82.09	2.66	0.07	0.56	0.16
Finish score ^d^	3.10	2.87	3.00	2.92	0.08	0.43	0.87	0.64
Back-fat thickness, mm	3.37	1.77	2.48	2.28	0.25	<0.01	0.38	0.52
Marbling score ^e^	4.95	5.48	5.20	5.36	0.15	0.03	0.66	0.80
Skeletal maturity ^f^	177.36	187.42	178.4	188.8	5.34	0.21	0.11	0.73
Lean maturity ^f^	188.42	214.83	196.0	213.6	8.89	0.04	0.07	0.269
Overall maturity ^f^	183.68	201.29	186.8.0	202.40	6.81	0.09	0.08	0.35
Adipose maturity ^g^	3.00	2.90	2.92	2.96	0.08	0.27	0.84	0.73
Carcass length, cm	130.32	131.84	130.58	131.84	3.51	0.06	0.11	0.46
Thigh width, cm	60.13	62.27	61.36	61.56	1.65	0.04	0.84	0.85
Length of pelvic limb, cm	54.97	57.12	57.21	55.41	1.58	0.62	0.16	0.57
Leg perimeter, cm	119.15	120.80	120.24	120.12	3.22	0.49	0.90	0.83
Thoracic depth, cm	36.58	38.30	37.32	37.99	1.09	0.64	0.88	0.49

^a^ ZER-ZER: corresponds to a dose (72 mg) of zeranol at d-0 followed by a second dose (72 mg) of zeranol at d-90. ^b^ TBA/E2-ZER: corresponds to a first dose containing 140 mg trenbolone acetate + 20 mg estradiol 17 beta at d-0, followed by a second dose (72 mg) of zeranol at d-90. ^c^ 1 = Very convex, 2 = Convex, 3 = Rectilinear, 4 = Concave, 5 = Very concave [28]. ^d^ where, 1 = Extremely abundant, 2 = Abundant, 3 = Medium 4 = Slight, 5 = Scarce [28]. ^e^ 1 = Abundant, 2 = Moderate, 3 = Small, 4 = Slight, 5 = Traces, 6 = Practically devoid [28]. ^f^ Carcasses within the 100–199 maturity range score represents the youngest group (100 is equal to A00 and 199 is equal to A99); 200–299: represent carcasses with intermediate, more advanced maturity (200 is equal to B00 and 299 is equal to B99) [3,44]. ^g^ 1 = Ivory white, 2 = Creamy white, 3 = Light yellow, 4 = Intense yellow, 5 = Orange [3].

**Table 8 animals-12-00366-t008:** Frequency distribution of carcass categories/grades according to male class and implant protocol (Experiment II).

Carcass Category/Grade	Male Class		Implant Protocol ^a^	
	Steer *n* (%)	Bull *n* (%)	ZER-ZER *n* (%)	TBA/E2-ZER *n* (%)
Venezuelan carcass category ^b^				
AA	8 (42.1)	0 (0.0)	5 (20)	3 (12)
A	4 (21.1)	3 (9.7)	4 (16)	3 (12)
B	5 (26.3)	21 (67.7)	13 (52)	13 (52)
C	2 (10.5)	7 (22.6)	3 (12)	6 (24)
	χ^2^ = 18.98; *p* = 0.002		χ^2^ = 1.64; *p* = 0.65	
USDA Carcass Quality Grades ^c^				
Select	4 (21.1)	0 (0)	3 (12)	1 (4)
High Standard	10 (52.6)	4 (12.9)	7 (28)	7 (28)
Low Standard	5 (26.3)	15 (48.4)	9 (36)	11 (44)
Bull	0 (0)	12 (38.7)	6 (24)	6 (24)
	χ^2^ = 21.96; *p* = 0.001		χ^2^ = 1.20; *p* = 0.75	
USDA Carcass Yield Grades ^d^				
1	6 (31.6)	15 (48.4)	10 (40)	11 (44)
2	2 (10.5)	0 (0)	1 (4)	1 (4)
3	11 (57.9)	16 (51.6)	14 (56)	13 (52)
	χ^2^ = 4.14; *p* = 0.12		χ^2^ = 0.080; *p* = 0.96	
Total	19	31	25	25

^a^ ZER-ZER: corresponds to a dose (72 mg) of zeranol at d-0 followed by a second dose (72 mg) of zeranol at d-90; TBA/E2-ZER: corresponds to a first dose containing 140 mg trenbolone acetate + 20 mg estradiol 17β at d-0, followed by a second dose (72 mg) of zeranol at d-90. ^b^ AA, A, B, and C Venezuelan carcass categories correspond to the first, second-, third-, and fourth-quality, respectively [3]. ^c^ Carcasses of bulls younger than 30 mo. of age and (or) exhibiting an A physiological maturity are designated in the “Bullock” class; USDA Standard quality grade corresponds to the fourth quality, for bullock carcasses [44]. ^d^ USDA yield grades (YG) are rated numerically, namely, 1, 2, 3, 4, and 5; a YG 1 carcass is expected to have the highest proportion (>53.5%) of boneless, closely trimmed retail cuts, while a YG 5 carcass is expected to have the lowest proportion (<44.3%) of boneless, closely trimmed retail cuts [44].

**Table 9 animals-12-00366-t009:** Carcass fabrication yield traits according to male class and implant protocol (Experiment II).

Variable	Male Class		Implant Protocol			*p*-Value		
	Steer (*n* = 19)	Bull (*n* = 31)	ZER-ZER ^a^ (*n* = 25)	TBA/E2-ZER ^b^ (*n* = 25)	SEM	CLASS	IMPL	CLASS × IMPL
High-valued boneless cuts,% ^c^	34.01	33.26	33.81	33.28	0.20	0.31	0.95	0.35
Medium-valued boneless cuts,% ^d^	23.66	25.16	25.07	25.31	0.32	<0.01	0.01	0.01
Low-valued cuts,% ^e^	21.53	21.33	21.53	21.29	0.21	0.29	0.27	0.28
Total clean bone,%	7.49	7.56	7.55	7.51	0.09	0.39	0.21	0.19
Trimmed fat,%	4.56	3.66	4.15	3.85	0.18	<0.01	0.02	<0.01
Total cuts,% ^f^	79.21	80.80	80.40	79.94	0.31	<0.01	0.17	0.05

^a^ ZER-ZER: corresponds to a dose (72 mg) of zeranol at d-0 followed by a second dose (72 mg) of zeranol at d-90; ^b^ TBA/E2-ZER: corresponds to a first dose containing 140 mg trenbolone acetate + 20 mg estradiol 17β at d-0, followed by a second dose (72 mg) of zeranol at d-90. ^c^ Tenderloin + ribeye and strip-loin + Top sirloin + Eye of round + Top (inside) round + Bottom (outside) round + knuckle + Tri-tip + Heel of round. ^d^ Shoulder clod and top blade + chuck tender + chuck roll. ^e^ Brisket +Inside skirt, flank, skirts + rib plate + shanks. ^f^ Total salable products consist of the sum of the high-, medium-, and low-valued cuts. Percentages were computed based on chilled carcass weight.

**Table 10 animals-12-00366-t010:** Meat quality and cookery traits according to male class and implant protocol (Experiment II).

Variables	Male Class		Implant Protocol			*p*-Value		
	Steers (*n* = 19)	Bulls (*n* = 31)	ZER-ZER ^a^ (*n* = 25)	TBA/E2-ZER ^b^ (*n* = 25)	SEM	CLASS	IMPL	CLASS × IMPL
Cooking loss,%	31.60	34.23	31.81	34.66	0.11	0.04	<0.01	0.24
Cooking time, min	77.75	78.87	77.30	79.62	0.64	0.58	0.82	0.16
WBSF, kg ^c^	5.52	5.50	5.29	5.74	0.29	0.11	0.04	<0.01
Muscle fiber tenderness ^d^	4.29	4.27	4.26	4.30	0.07	0.07	0.81	<0.01
ACT ^e^	3.78	3.75	3.76	3.75	0.08	0.87	0.96	<0.01
Overall tenderness ^d^	3.85	3.84	3.89	3.79	0.08	0.08	0.55	<0.01
Juiciness ^f^	4.98	4.97	4.89	5.07	0.06	0.05	0.06	0.14
Flavor intensity ^g^	5.97	5.74	5.76	5.93	0.05	0.02	0.08	0.72

^a^ ZER-ZER: corresponds to a dose (72 mg) of zeranol at d-0 followed by a second dose (72 mg) of zeranol at d-90; ^b^ TBA/E2-ZER: corresponds to a first dose containing 140 mg trenbolone acetate + 20 mg estradiol 17β at d-0, followed by a second dose (72 mg) of zeranol at d-90. ^c^ WBSF: Warner–Bratzler shear force. ^d^ 8-point hedonic scale, where 1 = extremely tough, and 8 = extremely tender. ^e^ ACT: Amount of connective tissue: 8-point hedonic scale, where 1 = abundant amount of connective tissue, and 8 = no connective tissue. ^f^ 8-point hedonic scale, where 1 = extremely dry, and 8 = extremely juicy. ^g^ 8-point hedonic scale, where 1 = extremely bland, and 8 = extremely intense.

## Data Availability

Data are not available in public datasets, please contact the authors.

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
