# Peer review of "The Effects of Castration, Implant Protocol, and Supplementation of Bos indicus-Influenced Beef Cattle under Tropical Savanna Conditions on Growth Performance, Carcass Characteristics, and Meat Quality"

_animals, 2022, doi:10.3390/ani12030366_

Round 1

Reviewer 1 Report

Report on paper no. animals-1547255

The effects of castration, implant protocol, and supplementation of Bos indicus-influenced beef cattle under tropical savanna conditions on growth performance, carcass characteristics, and meat quality

by Nelson Huerta-Leidenz , Nancy Jerez-Timaure * , Argenis Rodas-Gonzalez , Jhones Onorino Sarturi , Mindy M. Brashears , Markus F Miller , Michel Todd Brashears

Dear Authors

congratulations on your work, which is of both scientific and applicative interest, answering topics that are certainly of interest to ranchers.
it would be interesting to be able to evaluate under your conditions also the nutraceutical characteristics of the meat obtained in your farming systems (e.g. acid and vitamin profile...).

regards

General Comment

This paper addresses the important study area of managerial strategies to deal with forage shortages during seasonal droughts in pasture finishing of young cattle under tropical savanna conditions and able to improve carcass and meat quality, potentially increasing profitability and competitiveness. The paper adds some interesting information, addressing several aspects all of which are dealt with in a comprehensive manner. In general, the paper is relevant for the field and presented in a well-structured manner.

I recommend accepting the paper after a minor revision

General remarks:

I did not find any general remarks to make. Just a thought... why not use small letters in the tables?

Detailed remarks:

Line 48: “shortens feed the -the cycle of bulls, “…not clear.

Line 222: delete ; after the bracket

Line 300: “Non-significant male class × implant interactions”. being experiment 1 the interaction should be supplement X implant

Lines 350-352: the sentence starting with "in contrast..." seems to be a repetition of the previous sentence.

Line 588: delete a bracket

Figure 2 line 472: not SUPPL but MALE CLASS treatment

Reviewer 2 Report

Comment to Authors

This article describes research conducted on-farm in tropical Venezuela. The primary issue I have with the paper was also pointed out by the authors in the Conclusion, it is somewhat underpowered statistically to pick up some differences in growth and production responses. There is also a lack of balance in the experimental design. But this is explainable for on-farm research. For these management systems to be incorporated by producers in these underserved area research must be conducted in situ, so this effort should be congratulated. The meats work appears to be conducted well and the sample size seems to be appropriate for statistical analysis.

Forage quality and forage management should be described more fully. Also, the weather conditions during the experiments should be shown.

There are considerable grammatical errors in the Simple Summary and Abstract.

Line 19 reword to “Ranchers in tropical savannas must alter management to deal …”

Line 20 replace ‘and add’ with ‘while adding’

Line21 delete ‘assist in the pasture finishing of young cattle with’ and change ‘improve’ to ‘improved’

Line 22 Competitive against what? Please expound on this

Line 26 change to ‘In experiment II implant protocols were compared on bulls…’

Line 28 change ‘expensive meats’ to ‘high-value meat’

Line 33 delete ‘if it is bull or steer’ and replace with ‘castration status’

Line 33 replace operating with finishing calves

Line 35 what forages were utilized?

Line 43 change to ‘…and yield of high…’

Line 44 change to ‘…the time to harvest…’

Line 48 “feed the -the cycle of bulls” I do not follow what is meant here. Please reword

Line 56 reword to ‘with the expectation of compensatory gains’

Line 62 replace Stockers with Producers

Line65 delete ‘With’

Line 68 reword to ‘In confined feeding operations bulls outweigh…’

Line 71 reword to ‘…and strategic pasture supplementation have…’

Line 119 how many bulls and what were the ages?

Line 144 insert ‘was fed’ after ‘…Table 2)” to make this a complete sentence

Line148 “Climatic conditions allowed” what climatic conditions and what did the conditions allow?

Line 151 define “satisfactory conformation”

Line 567 replace ‘usually’ with ‘typically’

Line 572 delete ‘as a matter of fact’

Line 611 change to ‘…bulls finished on pasture…’

Line 664-665 change to ‘…the greater WBSF values (8.35 kg) than cattle fed forage only.”

Line 706 change to ‘In Experiment I…’

Reviewer 3 Report

This is a well written manuscript. The data presentation is simple and straightforward. The methods are explained well. The one issue I have is the experimental unit. I assume that individual animal was used as the experimental unit, but I am not sure that is appropriate.

Edits:

L23-25 - not sure you need to include specific treatments int he simple summary

L44 - "decreasing the to harvest", this phrase does not make sense to me

L48 - "shortens feed the -the cycle of bulls", this phrase does not make sense to me

L66 - I think this should be "technologies, it is possible"

L145 - change to feather meal

Table 1 - why were there more bulls than steers? Why for Senepol and Brahman did 1 treatment only 2 and the other 4 animals? Why not 3 and 3?

Table 1 footnote - change to "a dose (72 mg) of zeranol", check this is all tables as some time there is an extra "of" or the parentheses need to be removed

L168 - why 19 steers and 31 bulls?

L168-182 - were animals in Exp II individually fed? If not how do you ensure that all animals could eat the correct amount of supplement?

L175 - change to "Castrated (steer) and intact (bull) calves"

L251 - need to state the experimental unit

L254 - change to 'a linear mixed model'

L264 - remove 'Steel and Torrie' to be consistent with citations

Table 4 footnote - seems that 'extremely tough' and 'extremely tender' are repeated too many times

Table 6 - need units for ADG1

Table 9 - need units for the rows for 'cuts'

L519 - why the parentheses around 'of male class'?

L564 - change to 'supplement could'

L568 - do not need to say 'in both experiments' and list experiment I and experiment II

L581 - change 'trended' to 'tended' throughout manuscript

L588 - citation [48] is repeated

L603 - change 'remarked' to 'noted', in my opinion it reads better

L608 - delete word 'higher'

L631 - 'further reduce the expected, a lower', this phrase does not make sense

L646 - change 'experiences' to 'experiments'

L720-725 - this whole paragraph is 1 sentence. need to divide it up

Table S1 - the title should say 'Table S1'. why are number of animals in angus, romosinuano, and senepol not more even among treatments?

Table S2 - the title should say 'Table S2'. need to indicate whether these values are on dry matter or wet basis. under mineral there is a comma instead of a decimal point
